# A New 3D Cultured Liver Chip and Real-Time Monitoring System Based on Microfluidic Technology

**DOI:** 10.3390/mi11121118

**Published:** 2020-12-16

**Authors:** Yao Zhang, Ning Yang, Liangliang Xie, Fangyu Shu, Qian Shi, Naila Shaheen

**Affiliations:** 1School of Electrical and Information Engineering, Jiangsu University, Zhenjiang 212013, China; zhangyao20201031@163.com (Y.Z.); sfy15716109836@163.com (F.S.); Shiqian_0916@outlook.com (Q.S.); nailashaheen10@yahoo.com (N.S.); 2Faculty of Natural and Mathematics Science, King’s College London, Strand, London WC2R 2LS, UK; xieblink@163.com

**Keywords:** microfluidic technology, in vitro model of the liver, drug testing, near-infrared spectrum, electrochemical impedance spectrum

## Abstract

In vitro models of the liver have a good simulation of the micro-liquid environment inside the human liver and the communication between cell tissues, which provides an important research tool for drug research and liver disease treatment. In this paper, we designed a 3D liver chip and real-time monitoring system based on microfluidic technology. The in vitro model of the liver on the chip was established by the three-dimensional microfluidic chip pipeline and the corresponding microwell array. Meanwhile, the culture medium is continuously injected on the chip, and the electrochemical impedance spectroscopy and near-infrared spectroscopy of the liver chip are recorded and analyzed from day one to day five. When the 3D cultured liver chip in vitro model reached a certain period and stabilized, paracetamol with varying gradients of concentration was applied to the cultured cells for drug resistance testing. The experimental results show that the liver chip and its monitoring system designed in this paper can maintain 100% cell viability of hepatocytes in vitro for a long time. Furthermore, it can meet the requirements of measurement technologies such as electrical impedance measurement and near-infrared spectroscopy in real-time, providing a stable culture platform for the further study of organ chips.

## 1. Introduction

The liver is the largest parenchymal organ in the human body, which is responsible for various proteins, urea secretion and metabolism detoxification [1,2]. However, an unhealthy lifestyle can easily lead to liver damage, which may lead to liver failure or even be life-threatening if it is serious [3,4]. In the treatment of liver diseases and drug screening tests, the liver chip system can provide potentially useful information [5,6]. The liver chip system has a good simulation of the communication between the internal microenvironment and cell tissues of the human liver [7,8], can predict the effectiveness of drugs in the human body and provides an important research direction for the treatment of liver diseases [9]. Therefore, it is necessary to construct a liver chip system to provide a viable platform for accurate drug screening.

Due to the complex liquid microenvironment and microstructure of liver are complementary to microfluidic chips, which require less liquid, high throughput and integration [10,11], a microfluidic device is considered as a critical and reliable platform for drug screening research [12]. Therefore, scholars and organizations have conducted extensive research on liver chips based on microfluidic control in recent years [13,14,15]. Yosuke et al. proposed a microfluidic cell culture device that mimics the structure of liver tissue. The hepatocytes cultured in a two-dimensional double-row self-organize in a specific reactor and form a cord-like structure (hepatic cord), which realizes the function of the bile duct to the secretion of bile collection, and proves that a two-dimensional model can have better simulation to partial liver function [16]. In addition, Ho et al. formed different electric field strengths on the chip based on dielectrophoresis technology. The pearl chain structure formed by the mixed culture of hepatocytes and endothelial cells can better simulate the cord-like structure of the liver [17]. However, the limitation of two-dimensional culture makes it impossible for non-parenchymal cells such as liver cells and hepatic stellate cells to construct a microenvironment similar to that of cells in the human body. At the same time, the two-dimensional cultured tissue model and the human liver model have large physiological differences and physical environment differences, so it is impossible to accurately predict liver metabolism and drug screening. Although most researchers focus on the construction of liver models [18,19], more and more people realize that the real-time monitoring system of liver chips can quickly reflect the influence of drugs on the liver state, and is more conducive to accurate drug screening. Danny et al. allowed continuous measurement of the electrochemical data of glucose and lactate in the liver chip system through a computer-controlled microfluidic switchboard [20]. Zhang et al. integrated the physical sensor and electrochemical sensor with the liver chip to realize the automatic monitoring of the biophysical and biochemical parameters of the liver chip [21]. However, the monitoring time of these sensors is relatively short, and the integration of these sensors with the liver chip is difficult. The light transmittance of the chip is not fully utilized, and real-time and effective electrochemical and optical monitoring cannot be achieved.

Therefore, based on the above problems, this article proposes a 3D cultured liver chip and real-time monitoring system. Through the manufactured Polymethyl methacrylate(PMMA) chip and the multi-well plate integrated with the micro-well (300 μm) array embedded in the PMMA specific pipeline, the spheroid culture of hepatocytes is simulated to the greatest extent, and, as a result, the excellent reconstruction of the internal liver micro-tissue is achieved in vitro. The screen-printed electrodes integrated on the liver chip and the near-infrared spectroscopy measurement points meet the needs of directly recording and analyzing electrochemical data and spectral data on the chip, to realize the real-time monitoring of the electrochemical and optical parameters of the liver chip. Improving the accuracy of existing liver models in drug screening has led a new path.

## 2. Materials and Methods

### 2.1. Preparation of Cells

The liver cancer cell line (Hep-G2) was provided by the School of Pharmacy, Jiangsu University. The cell layer was rinsed with phosphate buffered saline (PBS) with pH 7.4. Then, the cells were separated using a trypsin-EDTA solution (0.25% trypsin and 1 mM EDTA·4Na; Gibco), and Dulbecco’s modified Eagle medium (DMEM; Gibco, Grand Island, NY) containing 10% fetal bovine serum (FBS) was added to the dispersed cell layer. The cell cultures were placed in a Likang Heal Force carbon dioxide incubator 160 W under a temperature of 37 °C and 5% CO_2_.

### 2.2. Development of Liver Chip Based on Microfluidic Technology

The liver chip was prepared by polymethyl methacrylate (PMMA) hot press molding technology [22]. The liver chip was designed and manufactured by Luzhi Heng Micro Precision Machinery Factory in Wuzhong District. The manufactured microfluidic liver chip consists of a plexiglass (PMMA) plate with microfluidic channels engraved on the lower layer, six porous plates and a PMMA plate with four through holes on the upper layer. The lower liver chip is shown in Figure 1a. The diameter of the liquid inlet was 1.5 mm, the etched pipes were 1mm deep, and the liquid storage pool was 2 mm deep. Behind the liquid storage pool was a screen-printed electrode connection with a height of 1mm and a width of 12 mm. At the same time, the multi-well plate embedding area had six pipes with a length of 3 mm and a width of 24 mm, which could well embed the porous culture plate with an etched pore size of 300 μm and had an excellent fixation effect on the printed liver polymer. The secondary detection area on the tablet shown was connected with two through holes in the upper liver chip with a diameter of 1.5 mm and a depth of 5 mm as the inlet and outlet of the region as shown in Figure 1b. If it is necessary to directly detect on the tablet through the introduction of peripheral reagents, the design of this area well meets the requirements. After adding an appropriate amount of culture medium to the lower liver chip, a layer of gelatin on the contact surface was applied, the upper and lower liver chips was aligned and pressure was applied to bond them together. The liver chip was completed as shown in Figure 1c. The schematic diagram of the multi-well plate and the printing of the three-dimensional cultured hepatocytes in its wells and the schematic diagram of the liquid flow are shown in Figure 1d, and each pore diameter was 300 μm. The printed liver spheroid aggregate was expected to be 200 μm. According to the study by Bhise et al. [23], hepatocyte spheroid aggregates of this size can not only prevent the necrosis of the central cell, but also maintain the liver function of the cells.

### 2.3. Establishment of Liver Chip Monitoring System

The established liver chip monitoring system based on microfluidic technology is shown in Figure 2, which was mainly composed of an injection pump, a CHI600E electrochemical workstation (Shanghai, China) and Ocean View Marine optical spectrometer (Hangzhou, China). At a flow rate of 200 μL/h, the DMEM medium for culturing liver cells was injected into the liver chip through the injection pump, and flowed out of the reservoir through the outlet of the liver chip into the collection dish. When a particular incubation time was met, the liver chip was connected to the spectrometer and electrochemical workstation to measure the spectrum and impedance. At the same time, paracetamol drug experiments with different concentration gradients were tested on the seventh day.

### 2.4. Experimental Method Based on Liver Chip Monitoring System

First, the chip sealing process and cell perfusion were performed. As shown in Figure 3, the used screen-printed electrode was placed in the reservoir of the bottom liver chip through the preset access port, and then it was placed in the reservoir with a liquid glue gun and glue stick (wiped with 75% ethanol) under ultraviolet light for five minutes to seal the inlet of the external screen-printed electrode. The extracellular matrix selected in the experiment was mainly gelatin and supplemented with other auxiliary materials. The cultured cells were mixed with the gelatin-based outer matrix to form a mixture of 1 × 10^7^ cells/mL, and 20 μL was drawn with a pipette. For the cell mixture, align the pipette tip to the wells of the multi-well plate for infusion one by one. In order to prevent the contact surface moisture from evaporating too fast and maintain the adsorption capacity of the liquid between the upper and lower layers of the liver chip, after the pretreatment, the appropriate medium was added to the bottom layer of the liver chip planted with cells, and a layer of gelatin was applied to the contact surface. After aligning the upper and lower layers of the chip, pressure was applied to fit and surround the chip with four layers of useful toughness bandages.

Then, according to the liver chip and monitoring system based on microfluidic technology as shown in Figure 2. At a flow rate of 200 μL/h, the DMEM medium for culturing liver cells was injected into the liver chip through a syringe pump and flowed out to the collection dish through the liver chip outlet, keeping 5% CO_2_ evenly and slowly passed into the incubator. At the same time, a blank cultured liver chip was placed in the incubator as a control group. It was injected at the same flow rate and was the same medium. When a certain incubation time was met, the liver chip and the blank control group were, respectively, connected to the spectrometer and electrochemical workstation to measure the spectrum and impedance. Configure the software in advance to shorten the experiment time and reduce pollution. After the liver cells were planted on the chip using a pipette, 12 h later, after the chip environment was stabilized, the first day’s spectral data and impedance data were recorded. Then the experiment was repeated until the fifth day.

At the same time, the liver chip was used for the cell drug experiment on the seventh day. Replace the culture medium with varying concentrations of paracetamol, the concentration gradients of which are 0, 1, 3, 5, 10 and 20 mM, respectively. After changing the culture medium, the liver cells were cultured in a Likang Heal Force carbon dioxide incubator (Guangzhou Ruike Biological Technology Co., Ltd., Guangzhou, China) 160 W for 4 h and stained with 0.4% Trypan Blue for 3 min. Then the drug resistance of the three-dimensional cultured spherical polymer of liver cells was observed through a microscope, and the viability of the cells was analyzed.

## 3. Results and Discussion

### 3.1. Results and Analysis of Electrochemical Impedance Spectroscopy

In this study, we measured impedance using three electrodes (working, reference and counter electrodes) to monitor cell growth in real time. After the chip was perfused with cells and placed in the cell incubator for 12 h, the screen-printed electrode was connected to the CHI600E electrochemical workstation for electrochemical impedance measurement. The initial settings were InitE (V) = 0.5 V, HighE (V) = 0.5 V, LowE (V) = −0.5 V and Sensitivity (A/V) was set to 1 × 10^−6^, and the scanning speed was 0.1 V/s. As shown in Figure 3, the impedance of the blank culture medium control chip, two-dimensional liver cells cultured in the clean dish, and three-dimensional spherical liver cells cultured in the designed liver chip were respectively monitored in real time from day one to day five. The results showed that as the number of days increased, the cells continued to grow. The impedance value of the cells under the three conditions became larger and the culture medium had the lowest impedance value, while the impedance value of the liver cells cultured on the liver chip exceeded that of two-dimensionally cultured liver cells on the fourth day, and stabilized on the fifth day.

As shown in Figure 4, on the first day of forward scanning, these three liquid medias all showed characteristic peaks around 0.1 V, while the impedance of the blank medium decreased rapidly near this voltage, followed by the two-dimensional culture medium. Further, the electrochemical resistance value tested by the liver chip drops to the slowest at this point, and the recovery time was also the fastest, about 1.2 s during forward scanning. When the voltage is 0 to 0.5 V, it can be observed that the chemical impedance of the reservoir based on the liver chip was the largest and had the smallest resistance peak. At the same time, the value of the two-dimensional cultured hepatocytes grown in the Petri dish is larger. The blank medium had the smallest resistance, which is opposite to the reverse scan data.

Measuring again on the third day, the initial settings were InitE(V) = 0.5 V, HighE(V) = 0.5 V, LowE(V) = −0.5 V and Sensitivity(A/V) was set to 1 × 10^−6^. The scanning speed was 0.1 V/s, from which the data shown in Figure 5 was obtained. It can be analyzed that the medium resistance of the hepatocytes derived from two-dimensional culture was the largest in the culture dish, followed by the data of the liver chip 3D culture. The resistance of the medium was the smallest. It can be inferred that the ion content in the solution increases through life activities in the experimental platform with cells cultured on the third day. The two-dimensional cells inoculated in the Petri dish on the third day were more active, because the liver cells cultured in 3D spherical aggregates on the liver chip need more time to grow and be stable to show stronger liver function than the two-dimensional cultured liver cells.

### 3.2. Results and Analysis of Near Infrared Spectroscopy

In order to further confirm that the liver cells cultured on the liver chip have stronger liver functions, the liver cells cultured on the liver chip and the control group were analyzed by near infrared spectroscopy. For the control group, the model was the same as the liver chip, but the porous cell plate did not contain printed liver cells and the liquid flow rate of the syringe pump was the same as the liver chip. Near-infrared spectrum data were recorded on the first day and the fifth day of culture for the control group with the only medium, respectively, which has the most considerable difference at the wavelength of 2380 nm as shown in Figure 6. Similar rules were observed in the 3D liver chip’s near-infrared spectra, but there was still no significant difference in the near-infrared band. The first day of the control group’s spectrum is located in the 2380 nm band with a transmittance of 243 and a value of 82 at the small peak (1890 nm). On the fifth day, the 2380 nm band’s value was 141 and the corresponding transmittance of 1890 nm was 57. On the first day, the value of the 3D liver chip in the 2380 nm band was 252, the small peak (1890 nm) was 85 and the value of the two bands was 114 and 35, respectively, on the fifth day. The light transmittance of these two bands for the control group decreased to a certain extent, but the data is not obvious. It is planned to use the absorbance spectrum for follow-up test observation.

The absorbance measured from the first day to the fifth day with the wavelength of 1950 nm and the comparison between the control group and the 3D liver chip was shown in Figure 7. The absorbance of the control group decreased with the increase of days, while the absorbance measured in the 3D liver chip reached the lowest on the third day and increased with the increase of days and stabilized on the fifth day. As shown in Figure 8, it can be found that the medium has a high absorption value in the wavelength range of 700 to 800 nm, and this feature can also be used in the 3D liver chip. It can be concluded that the main cause of this part of the change was the nutrients in the medium. At the same time, it can be observed that in the wavelength range of 1000 nm to 2800 nm, the absorbance value of the data measured on the 3D liver chip on the third day was generally lower than that measured on the first day. The cell condition was not good when the cells were printed on the liver chip on the first day, but there were still some secretions and metabolites between the cells and the medium, so the absorbance spectrum of this band was slightly higher than that on the third day. After printing, the liver cells slowly recover their ability to grow and proliferate on the chip, and gradually enhance the simulation of the in vivo microenvironment. Therefore, the absorption value of the corresponding wavelength band on the fifth day was more significant than that of the liver chip on the third day. There were two absorption peaks at 1950 nm and 2640 nm, and the absorption values were 0.323132 and 0.38306, respectively. The absorption spectrum data observed on the first day, the data of the control group and 3D liver chip were the same. This situation should also prove the sudden change of the cell culture environment on the first day of printing cells. Therefore, it is difficult for cells to quickly restore normal cell activity and physiological cell morphology, so the experimental data of the control group and the experimental chip group have fewer fluctuations. Therefore, the designed liver chip can meet normal liver cells’ requirements to grow, proliferate and maintain certain life activities in vitro. Simultaneously, it can perform electrical impedance and spectral data recording on the chip, so that microfluidic technology can be used to design an organ chip research platform with high integration, high sensitivity and good simulation of human microcirculation.

### 3.3. Results and Analysis of Paracetamol Metabolism

Figure 9 shows the cell viability test after four hours of culture in DMEM medium containing different drug concentration gradients. 

The concentration increases from left to right from top to bottom. When cultured with 0 mM paracetamol for four hours, it can be observed that the cultured liver cells were all translucent, maintain good morphological characteristics and have a high survival rate. After culturing with 1 mM paracetamol for four hours, a few dead cells can be seen in the sampling area that is stained dark blue, the number of apoptotic cells is still small, and the cell membranes of dead cells lose their inherent selective permeability. So, it will be stained by foreign biological stains. The subsequent 3 mM, 5 mM paracetamol cell culture medium, and samples with lower concentration gradients changed linearly, and the cells stained blue by 0.4% trypan blue stain gradually increased. When the cultured paracetamol concentration reaches 10 mM, a significant increase in apoptotic cells can be observed, which can prove that the drug concentration has exceeded the metabolic range that the liver cells can withstand. The internal microenvironment of the cultured liver cells has been destroyed. At this time, the number of surviving cells has significantly been reduced. At a concentration of 20 mM, it can be seen that it is challenging to observe unstained cells, and the cells undergo a large area of apoptosis. Therefore, as the concentration of paracetamol increases, the number of apoptotic cells also increases. At the same time, there are mutation points in the range of 10 mM, and according to the activity test of cells cultured at a concentration of 20 mM, it can be seen that the metabolic capacity of liver cells cultured on the liver chip under study is about 10 mM.

## 4. Conclusions and Discussion

In this paper, a new type of liver chip and real-time monitoring system based on microfluidic technology is designed. The establishment of an in vitro liver model is achieved through the manufactured PMMA chip and a multi-well plate with an integrated microwell (300 μm) array embedded in a specific PMMA pipeline. Simultaneously, the medium is continuously injected on the chip, and the impedance value and spectral value at different time points are measured by the electrochemical workstation and the near-infrared spectrometer. After the 3D cultured liver chip in vitro model reached a certain period and was stable, the cultured cells were applied with different concentration gradients of paracetamol for drug experiments. Experimental results show that the designed liver chip can still ensure a good survival rate of liver cells and a good proliferation after being cultured in a flowing medium for seven days. The performance of three-dimensional cultured liver cells in the short term may be slightly inferior to that of two-dimensional cultured cells. Still, as the culture time increases, they will also show better liver functions in vitro. The liver chip can directly perform electrochemical and optical analysis on the outside of the chip without affecting the internal environment of the chip.

Although the liver chip designed in this paper has the stability and reliability of culture and can collect optical and electrical impedance information on the chip, as the culture time increases, a certain amount of leakage occurs after 10 days, making the experiment unable to continue. Therefore, PMMA sheet can be added to the upper and lower layers of the chip. The upper and lower layers of the chip and the edge of the PMMA sheet are integrated with nut holes. Some bio-affinity glue is applied to the contact surface, and then the entire chip is fixed by the nut, thus better improve of the performance of the chip.

## Figures and Tables

**Figure 1 micromachines-11-01118-f001:**
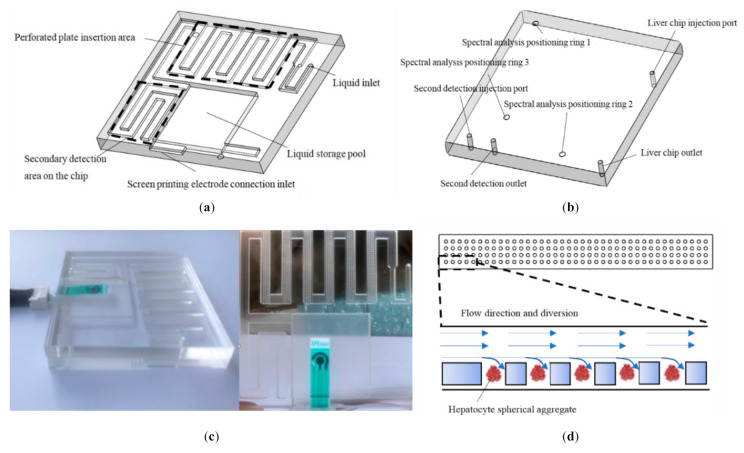
View of each part of the liver chip: (**a**) schematic diagram of the lower liver chip; (**b**) schematic diagram of the upper liver chip; (**c**) fabricated liver on a chip; and (**d**) porous pad and its schematic diagram of liquid-flow.

**Figure 2 micromachines-11-01118-f002:**
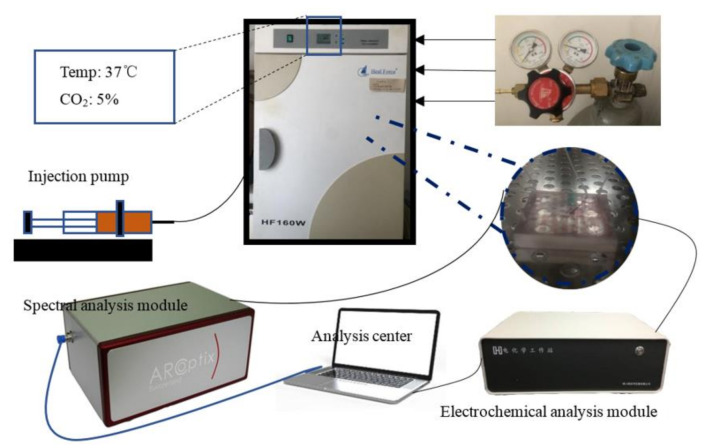
Schematic layout of the liver-on-chip and its analysis system.

**Figure 3 micromachines-11-01118-f003:**
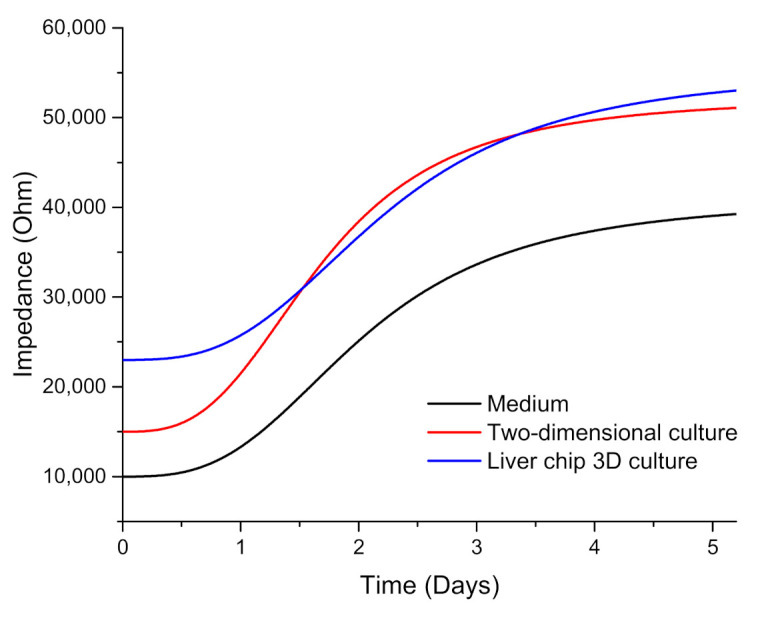
Impedance monitoring in different culture environment.

**Figure 4 micromachines-11-01118-f004:**
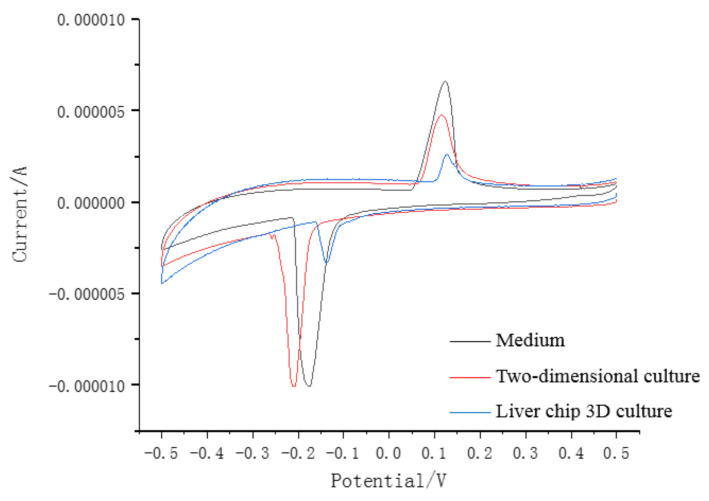
The electrochemical impedance of various liquid environment (day 1).

**Figure 5 micromachines-11-01118-f005:**
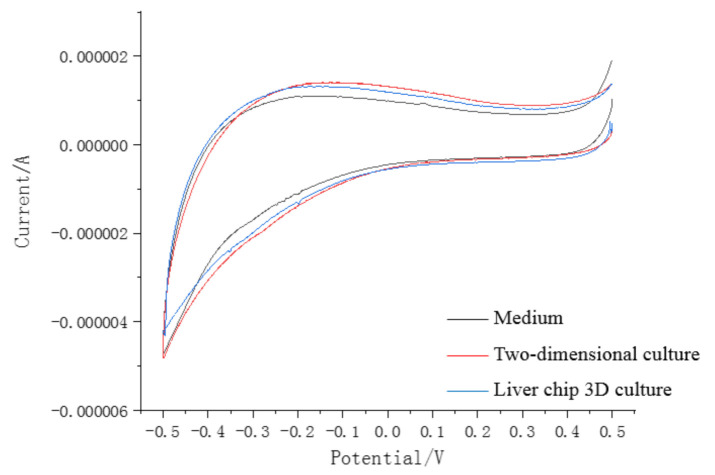
The electrochemical impedance of various liquid environment (day 3).

**Figure 6 micromachines-11-01118-f006:**
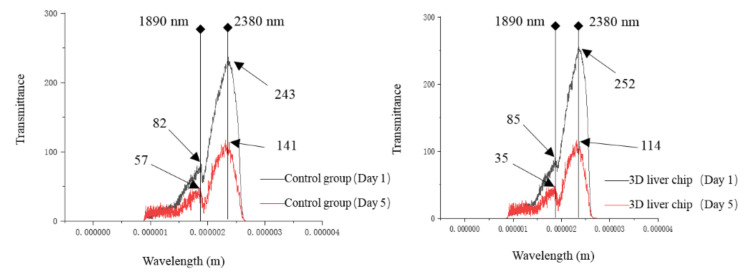
Results of near-infrared spectroscopy.

**Figure 7 micromachines-11-01118-f007:**
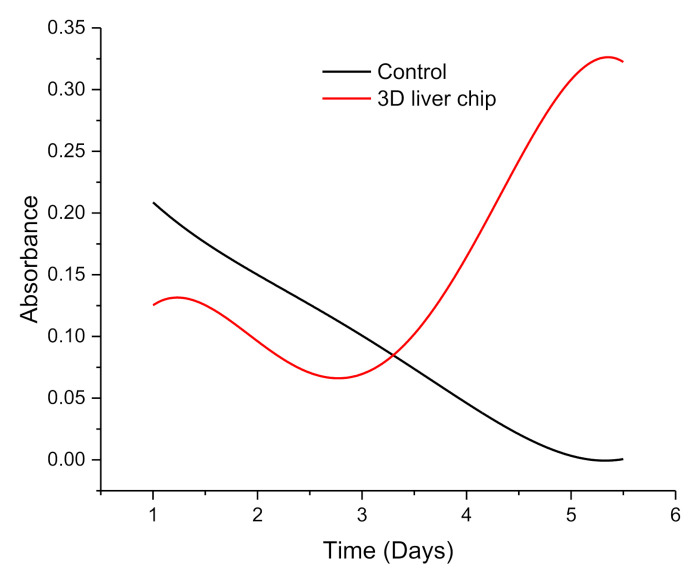
Absorbance monitoring during liver cell culture.

**Figure 8 micromachines-11-01118-f008:**
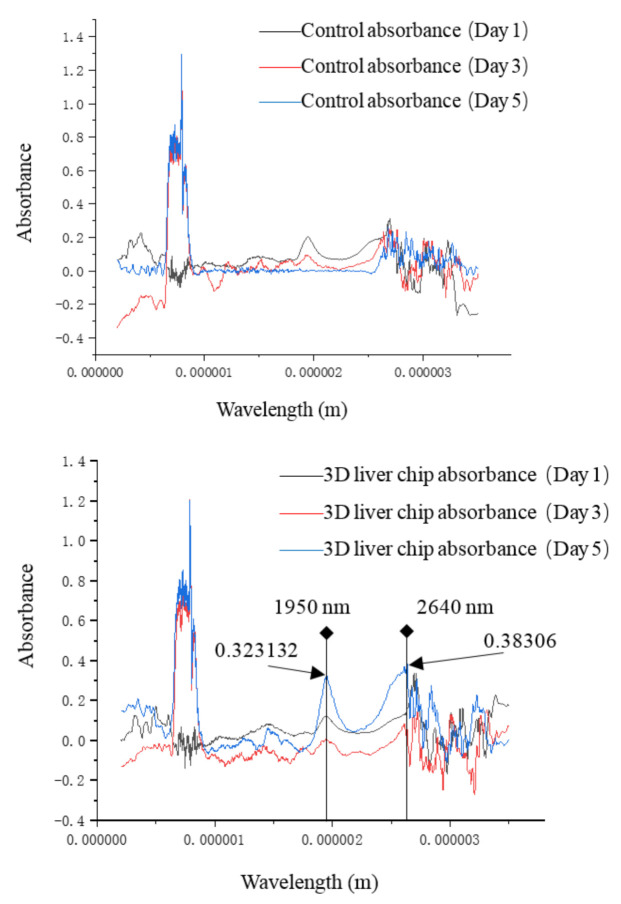
Results of absorbance.

**Figure 9 micromachines-11-01118-f009:**
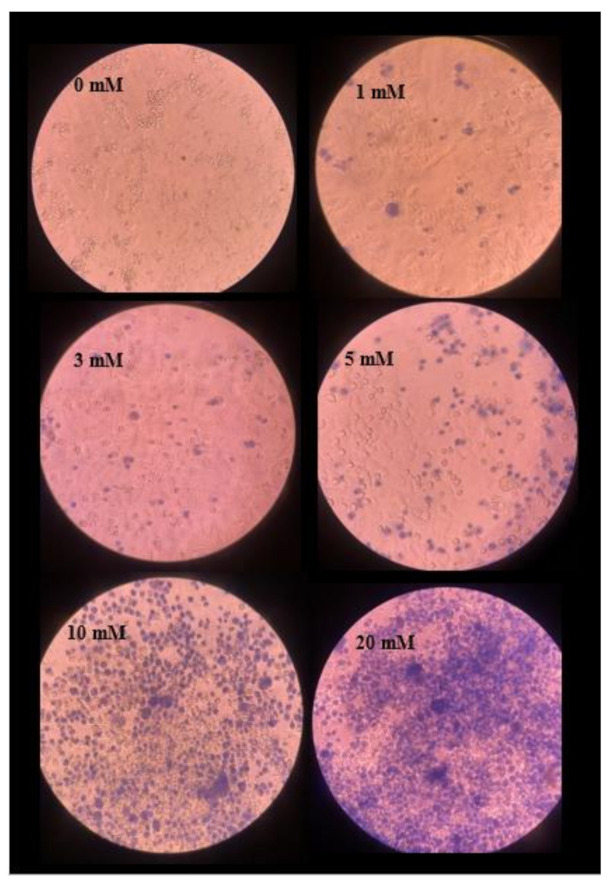
Results of cell viability in various concentration of paracetamol.

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
