# Peer review of "A New 3D Cultured Liver Chip and Real-Time Monitoring System Based on Microfluidic Technology"

_micromachines, 2020, doi:10.3390/mi11121118_

Round 1

Reviewer 1 Report

A New 3D Cultured Liver Chip and Real-Time Monitoring System Based on Microfluidic Technology

The presented manuscript aims the development of a 3D liver chip with real-time monitoring system based on microfluidic technology. Although the stated interest of the proposed liver chip for drug screening tests, the work lacks on novelty and significancy. There is nothing new or relevant in this study. The study was poorly conducted, no relevant data is shown, and the authors claim a real-time monitoring system that has to be taken from the incubator to be read! How this is suitable for continuous and real-time monitoring of cells?

Additionally, the manuscript needs an extensive editing on English language.

Therefore, the manuscript has no interest to the readers of Micromachines journal and should be rejected.

Author Response

Dear Reviewer,

We have studied the valuable comments carefully, and we will try our best to revise the manuscript (micromachines-999590). The point to point responses to your comments are listed as follows:

Reviewer 1

Comment 1: The presented manuscript aims the development of a 3D liver chip with real-time monitoring system based on microfluidic technology. Although the stated interest of the proposed liver chip for drug screening tests, the work lacks on novelty and significancy. There is nothing new or relevant in this study. The study was poorly conducted, no relevant data is shown, and the authors claim a real-time monitoring system that has to be taken from the incubator to be read! How this is suitable for continuous and real-time monitoring of cells?

Response: Thank you for your valuable comment.

(1) Because the traditional two-dimensional cultured tissue model cannot reconstruct the liver functional microtissues in vitro and cannot monitor the metabolic process of liver cells in real time. Therefore, this paper designs a 3D cultured liver chip and real-time monitoring system. The screen-printed electrodes integrated on the liver chip and the measurement points of the near-infrared spectroscopy meet the requirements of directly recording and analyzing electrochemical data and spectral data on the chip. In order to realize the real-time monitoring of the electrochemical and optical parameters of the liver chip, it has led a new way to improve the accuracy of the existing liver model in drug screening.

(2) This article adds electrochemical impedance and optical data from day 1 to day 5 under three different conditions in the results section and some clear descriptions, showing the real-time monitoring of impedance and optical during liver cell culture. The cell growth of the cell chip can be monitored more conveniently and continuously (please see Figure 4 and line 164-170; Figure 8 and line 223-227).

(3) In our experiment, the liver chip was taken out of the incubator connected with the injection pump after a certain culture time was satisfied, and the spectra and impedance were measured by connecting with the spectrometer and electrochemical workstation respectively. The liver chip was removed from the incubator. At this time, the liver cells in the liver chip can maintain 100% cell viability of hepatocytes in vitro for a long time. The real-time monitoring effect of the cells could be achieved by connecting with the spectrometer and electrochemical station respectively

Comment 2: Additionally, the manuscript needs an extensive editing on English language.

Response: We tried our best to improve the manuscript and made some changes in the manuscript. These changes will not influence the content and framework of the paper. And here we did not list the changes but marked in red at the revised paper. We appreciate for your warm work earnestly, and hope that these adjustments will meet with approval.

Comment 3: The manuscript has no interest to the readers of Micromachines journal.

Response: The research content of the journal "Micromachines" involves micro/nano structures, materials, devices, systems, and basic research and applications related to micro and nano technology. In this paper, a real-time monitoring system based on the liver chip is designed to realize the real-time monitoring of the electrochemical and optical parameters of the liver chip, which is in line with the research content of the "Micromachine" journal.

Reviewer 2 Report

The authors have developed a microfluidic chip for real-time monitoring of 3D cultured liver cells. The application of near-infrared spectrum and electrochemical impedance spectrum for monitoring the cell status of health is very interesting and useful. But there are major problems with this work:

  1. The microfluidic chip and the assembly of the lower and upper liver chip are not clear. One single figure combing figures 1 and 2 in addition to a new schematic for assembly can help to understand.
  2. The results section demonstrates electrochemical impedance for day 1 and day 5 in 3 different conditions. We don't know what happens between days 1 and 5. The authors should define an index out of this spectrum and report the measurements for e.g. 5 days in a certain interval.
  3. The results section demonstrates near-infrared spectroscopy for a range of wavelengths on 1, 3, and 5 days. Same to the electrochemical impedance measurements, the authors should come up with an index as a representative of this measurement and report the changes that happen during these 5 days in a certain interval.
  4. In section 3 of the results, the analysis of paracetamol metabolism is reported. However, the authors are not using the two developed approaches in the previous 2 sections (near-infrared spectrum and electrochemical impedance spectrum) for reporting the drug analysis. This section of the results should be completed using the developed approaches.
  5. The second paragram of the conclusion should be moved to the discussion section.

Author Response

Dear Reviewer,

We have studied the valuable comments carefully, and we will try our best to revise the manuscript (micromachines-999590). The point to point responses to your comments are listed as follows:

Reviewer 2

Comment 1: The microfluidic chip and the assembly of the lower and upper liver chip are not clear. One single figure combing figures 1 and 2 in addition to a new schematic for assembly can help to understand.

Response: Thank you for your useful comment. We have added a new schematic for the assembly of the lower and upper liver chip and updated the corresponding descriptions (please see Figure 1c and line 91-93) and hope that these adjustments will meet with approval.

Comment 2: The results section demonstrates electrochemical impedance for day 1 and day 5 in 3 different conditions. We don't know what happens between days 1 and 5. The authors should define an index out of this spectrum and report the measurements for e.g. 5 days in a certain interval.

Response: We have added the electrochemical impedance at days 1 to day 5 under three different conditions and reported the measurements (please see Figure 4 and line 164-170).

Comment 3: The results section demonstrates near-infrared spectroscopy for a range of wavelengths on 1, 3, and 5 days. Same to the electrochemical impedance measurements, the authors should come up with an index as a representative of this measurement and report the changes that happen during these 5 days in a certain interval.

Response: We have added the near-infrared spectroscopy from day 1 to day 5 and reported the measurements (please see Figure 8 and line 223-227).

Comment 4: In section 3 of the results, the analysis of paracetamol metabolism is reported. However, the authors are not using the two developed approaches in the previous 2 sections (near-infrared spectrum and electrochemical impedance spectrum) for reporting the drug analysis. This section of the results should be completed using the developed approaches.

Response: We agree with your significant comment. Although liver chip design in this paper with foster stability and reliability, as well as on optical and electrical impedance information collection, but as the growth of the incubation time there will be some leakage, so using two section before the development of two methods (near infrared spectrum and electrochemical impedance spectroscopy) report pharmaceutical analysis can't achieve at present, the subsequent we will optimize the structure of the chip and do further analysis of the experiment.

Comment 5: The second paragram of the conclusion should be moved to the discussion section.

Response: Thank you for your meaningful comment. We have changed the part of conclusions to conclusions and discussion (please see line 282) and hope that these adjustments will meet with approval.

Reviewer 3 Report

This paper presents a 3D microfluidic-type chip for culturing liver cells as an attempt to develop an in vitro liver model. The results presented and the conclusions drawn based on the findings have merit. Testing is achieved using electro-chemical impedance contrasting 2D and just medium over several days. NIR is also used to test the conditions and health of the cells in the 3D liver chips over several days, based on absorbance spectra. The approach is finally tested for the viability of the cells using different concentrations of paracetamol. 

The paper is presented well, overall, but there are some important aspects to take care of. Please find the comments below.

- A general revision of the style and language in the paper is required.

- The results subsections require a paragraph, or perhaps 1-3 sentences, introducing the subsection and the reasons and importance of the techniques used for the characterization and testing of the chips.  Section 3.2, for instance, starts immediately describing the control group. It would be good, for the readership, to include a justification on why NIR was used and the context of the experiments.

- The information provided for the preparation of Hep-G2 cells is not sufficient. The authors should provide information on the actual preparation of cells (as per the section title) in addition to mentioning that they were provided by the School or Pharmacy. 

- Provide a reference for "hot press molding technology" referred in Section 2.2. for the fabrication of PMMA microfluidic chips.

- Figures 1 -3 can be bundled into one single figure.

Author Response

Dear Reviewer,

We have studied the valuable comments carefully, and we will try our best to revise the manuscript (micromachines-999590). The point to point responses to your comments are listed as follows:

Reviewer 3

Comment 1: A general revision of the style and language in the paper is required.

Response: We have found a native English speaker from America to help us revise the whole text (please see the phrase in red).

Comment 2: The results subsections require a paragraph, or perhaps 1-3 sentences, introducing the subsection and the reasons and importance of the techniques used for the characterization and testing of the chips. Section 3.2, for instance, starts immediately describing the control group. It would be good, for the readership, to include a justification on why NIR was used and the context of the experiments.

Response: We have added reasons and experimental background for using impedance and near-infrared spectroscopy measurements in the section of the results (please see line 159-160 and line 203-205).

Comment 3: The information provided for the preparation of Hep-G2 cells is not sufficient. The authors should provide information on the actual preparation of cells (as per the section title) in addition to mentioning that they were provided by the School or Pharmacy. 

Response: We have added details about the actual preparation of cells (please see line 72-75).

Comment 4: Provide a reference for "hot press molding technology" referred in Section 2.2. for the fabrication of PMMA microfluidic chips.

Response: We have added a reference for "hot press molding technology" referred in Section 2.2. for the fabrication of PMMA microfluidic chips (please see line 78).

Comment 5: Figures 1 -3 can be bundled into one single figure.

Response: We have bundled Figures 1 to Figure 3 into one single figure (please see Figure 1).

Reviewer 4 Report

The paper by Zhang et al, presents a microfluidic device for the culturing of hepatocytes and measurement using electrical impedance and near-infrared spectroscopy. Overall the manuscript has potential to present some interesting research but there are a number of areas which require improvement before publication can be recommended.

General:

In vitro should be in italics throughout.

The image quality of most of the figures is poor and resolution needs improving for the reader. The figure legends are very sparse and need much more detailed information in them.

The level of methodological detail in parts is too much for a manuscript, e.g. turning blue knob on the tank (lines 147/8), line 169 etc.

Abstract:

Line 14 - It is unclear at the beginning what the ‘precarved three-dimensional microfluidic chip pipeline’ is. Please explain more clearly.

Line 17 - Can you add in the time points / range?

Line 18  - How do the paracetamol concentrations relate to clinical levels? This should also be explained more clearly throughout, particularly in the results and discussion section.

Line 21 - How long is ‘a long time’?

Introduction:

Paragraph 1 – Use of ‘the’ live chip system implies the one presented here but I think the introduction is more general and should refer to ‘a’ liver chip system.

Line 39 – Please remove the term ‘at home and abroad’ as the audience is international.

Paragraph 2 – full names of other authors are not required.

Materials and methods:

Line 87 - Check dimension of thickness 300 m?

Figure 1 – 3 should be combined into a single figure with multiple parts. A scale bar should be included. Clearly label where the porous pad is located within the main chip image.

Figure 4 – Is this figure really necessary, it doesn’t feel like it adds a lot to the manuscript.

Figure 5 – What is the purpose of the left hand side image? It would be better to include the photograph of the chip being loaded larger and with a scale.

Integration of spectrophotometer and impedance measurements is unclear and requires more detail.

Line 162 - How relevant are the paracetamol levels in terms of clinical levels?

Results and Discussion:

Impedance spectroscopy  - please explain why there is more variation in the negative peaks for the different samples? Were the cultures measured on any other days other than 1 and 5?

Infrared – no real discussion of what features are shown.

Paracetamol metabolism – were the results in Fig 10 from cells on the chip? Were they photographed in situ? Needs scale bar. How does this paracetamol concentration relate to clinical levels? How do you view the full 3D structure properly to look at dead cells rather than just slices through, is this truly representative?

Conclusion:

Line 288 - How are you defining ‘good’ survival and ‘good’ proliferation?

Line 289 – You talk about proliferation for 7 days but earlier results are only presented for 5 days.

Author Response

Dear Reviewer,

We have studied the valuable comments carefully, and we will try our best to revise the manuscript (micromachines-999590). The point to point responses to your comments are listed as follows:

Reviewer 4

Comment 1: General: In vitro should be in italics throughout. The image quality of most of the figures is poor and resolution needs improving for the reader. The figure legends are very sparse and need much more detailed information in them. The level of methodological detail in parts is too much for a manuscript, e.g. turning blue knob on the tank (lines 147/8), line 169 etc.

Response: Thank you for your valuable comment. We have change all in vitro in the article to italics and marked in red. We have made most modifications to the pictures of the article and hope to get your approval. We have partially integrated the images and added some descriptions. We have removed some methodological details (please see line 142 and line 160-162).

Comment 2: Abstract: Line 14 - It is unclear at the beginning what the ‘precarved three-dimensional microfluidic chip pipeline’ is. Please explain more clearly. Line 17 - Can you add in the time points / range? Line 18 - How do the paracetamol concentrations relate to clinical levels? This should also be explained more clearly throughout, particularly in the results and discussion section. Line 21 - How long is ‘a long time’?

Response: Thank you for your useful comment. The precarved three-dimensional microfluidic chip pipeline refers to the pre-designed three-dimensional microfluidic chip pipeline. We have added in the time range (please see line 17). Paracetamol is one of the most commonly used analgesics worldwide, and overdoses are associated with lactic acidosis, hepatocyte toxicity, and acute liver failure due to oxidative stress and mitochondrial dysfunction. We consulted many literatures, such as analytical chemistry journals, and selected paracetamol with a concentration gradient of 5,10.20 mM, while the cell activity at 20 mM was only about 40%. Therefore, we selected paracetamol with a concentration gradient of 0,1.3, 5,10,20 mM for drug experiments. A long time can last for about 10 days, which is explained in the conclusions and discussion.

Comment 3: Introduction: Paragraph 1 – Use of ‘the’ liver chip system implies the one presented here but I think the introduction is more general and should refer to ‘a’ liver chip system. Line 39 – Please remove the term ‘at home and abroad’ as the audience is international. Paragraph 2 – full names of other authors are not required.

Response: Thank you for your meaningful comment. There are many liver chip systems, so ‘the’ liver chip system is used to represent a wide range. In this paper, there is only one kind. We have remove the term ‘at home and abroad’ (please see line 39). We have modified the author information and did not display full the author names (please see the phrase in red).

Comment 4: Materials and methods: Line 87 - Check dimension of thickness 300 m? Figure 1 – 3 should be combined into a single figure with multiple parts. A scale bar should be included. Clearly label where the porous pad is located within the main chip image. Figure 4 – Is this figure really necessary, it doesn’t feel like it adds a lot to the manuscript. Figure 5 – What is the purpose of the left hand side image? It would be better to include the photograph of the chip being loaded larger and with a scale. Integration of spectrophotometer and impedance measurements is unclear and requires more detail. Line 162 - How relevant are the paracetamol levels in terms of clinical levels?

Response: Thank you for your thoughtful comments. Maybe the details were not checked properly at that time. The thickness should be 300 um, which has been modified in the article. We have integrated Figures 1 through 3 into a single figure, clearly marking the location of the perforated plates (please see Figure 1). Figure 4 shows the liver chip monitoring system, which can help readers better understand how the system is built. Figure 5 shows a detailed diagram of the experiment to help the reader understand. Paracetamol is one of the most commonly used analgesics worldwide, and overdoses are associated with lactic acidosis, hepatocyte toxicity, and acute liver failure due to oxidative stress and mitochondrial dysfunction.

Comment 5: Results and Discussion: Impedance spectroscopy - please explain why there is more variation in the negative peaks for the different samples? Were the cultures measured on any other days other than 1 and 5? Infrared – no real discussion of what features are shown. Paracetamol metabolism – were the results in Fig 10 from cells on the chip? Were they photographed in situ? Needs scale bar. How does this paracetamol concentration relate to clinical levels? How do you view the full 3D structure properly to look at dead cells rather than just slices through, is this truly representative?

Response: We have redone the experiment and measured the impedance from day 1 to day 5 and some clear descriptions (please see Figure 4 and line 164-170). We have added some features in Section 3.2(please see line 203-205). Paracetamol metabolism - The results in Figure 10 are from the cells on the chip, which were photographed in situ. Paracetamol is one of the most commonly used analgesics worldwide, and overdoses are associated with lactic acidosis, hepatocyte toxicity, and acute liver failure due to oxidative stress and mitochondrial dysfunction. We consulted many literatures, such as analytical chemistry journals, and selected paracetamol with a concentration gradient of 5,10.20 mM, while the cell activity at 20 mM was only about 40%. Therefore, we selected paracetamol with a concentration gradient of 0,1.3, 5,10,20 mM for drug experiments. Although liver chip design in this paper with foster stability and reliability, as well as on optical and electrical impedance information collection, but as the growth of the incubation time there will be some leakage, so using two section before the development of two methods (near infrared spectrum and electrochemical impedance spectroscopy) report pharmaceutical analysis can't achieve at present, the subsequent we will optimize the structure of the chip and do further analysis of the experiment.

Comment 6: Conclusion: Line 288 - How are you defining ‘good’ survival and ‘good’ proliferation?

Line 289 - You talk about proliferation for 7 days but earlier results are only presented for 5 days.

Response: Good survival and good proliferation are conditions in which cells reach a stable stage. By monitoring the impedance and absorbance of the liver cells from day 1 to day 5, we found that the cells stabilized at day 5. On the seventh day, the drug experiment was carried out and cultured with 0 mM paracetamol for four hours. It was observed that the cultured liver cells were all transparent and maintained good morphological characteristics with a high survival rate. This indicated that the cells at day 7 had a good survival.

Round 2

Reviewer 1 Report

A New 3D Cultured Liver Chip and Real-Time Monitoring System Based on Microfluidic Technology

Although the authors have improved the content of the manuscript, this work stills poor and without novelty. The authors claim the integration of sensors for real-time monitoring, but this was not demonstrated in the article. Indeed, the microfluidic chip has to be taken from the incubator and read using bench equipment that are laborious, time consuming and prompt to contaminations. There are in literature many works that have developed 3D liver-organoid microfluidic devices with exceptional biological characterization of the organoids (not shown in the manuscript), namely viability, immunostaining and molecular studies, reporting important findings. Some examples:

  • A liver-on-a-chip platform with bioprinted hepatic spheroids, Bhise et al., 2016. DOI:10.1088/1758-5090/8/1/014101   
  • Scaffold-Free Liver-On-A-Chip with Multiscale Organotypic Cultures, Weng et al., 2017. DOI:10.1002/adma.201701545
  • Reproducing human and cross-species drug toxicities using a Liver-Chip, Jang et al., 2019. DOI: 10.1126/scitranslmed.aax5516

Other articles that also included the integration of biosensors:

  • Multisensor-integrated organs-on-chips platform for automated and continual in situ monitoring of organoid behaviors, Zang et al., 2017. DOI: 1073/pnas.1612906114
  • Real-time physiological sensor-based liver-on-chip device for monitoring drug toxicity, Farooqi et al., 2020. DOI:10.1088/1361-6439/ababf4

It is my expectation that the author could took the abovementioned works into account and improve their work. The manuscript needs at least the characterization of the organoids and the improvement of the sensor integration for a real-time monitoring.  Overall, the article should not be considered for publication.

Author Response

Reviewer #1

Comment 1: Although the authors have improved the content of the manuscript, this work stills poor and without novelty.

Response: Thank you for your significant comment. Through reading a lot of literature, I can see that there are relatively few articles on 3D liver chip, and there is no integration of electrochemical sensor and optical sensor with liver chip, and real-time electrochemical and optical monitoring of liver cells has not been realized.

Comment 2: The authors claim the integration of sensors for real-time monitoring, but this was not demonstrated in the article.

Response: Thank you for your valuable comments. In the results and discussion section of this paper, the electrochemical and optical parameters of 3D liver chips, 2D liver chips and petri dishes were monitored in real time from day 1 to day 5, and the data showed that 3D liver chips had better characteristics (please see sections 3.1 and 3.2).

Comment 3: Indeed, the microfluidic chip has to be taken from the incubator and read using bench equipment that are laborious, time consuming and prompt to contaminations.

Response: Thank you for your careful comments. The microfluidic chip was removed from the incubator and was time-consuming and susceptible to contamination, which we did not take into account. However, our experimental environment was sterilized at that time, so we modified this procedure in order to cause misunderstanding (please see line 112-113).

Comment 4: There are in literature many works that have developed 3D liver-organoid microfluidic devices with exceptional biological characterization of the organoids (not shown in the manuscript), namely viability, immunostaining and molecular studies, reporting important findings. Some examples:

  • A liver-on-a-chip platform with bioprinted hepatic spheroids, Bhise et al., 2016. DOI:10.1088/1758-5090/8/1/014101
  • Scaffold-Free Liver-On-A-Chip with Multiscale Organotypic Cultures, Weng et al., 2017. DOI:10.1002/adma.201701545
  • Reproducing human and cross-species drug toxicities using a Liver-Chip, Jang et al., 2019. DOI: 10.1126/scitranslmed.aax5516

Response: Thank you for your thoughtful advice. They monitored liver function through the concentration of secretions. In this paper, impedance and absorbance were used to characterize liver function. When liver function was stable, drug toxicity analysis was performed.

Comment 5: Other articles that also included the integration of biosensors:

  • Multisensor-integrated organs-on-chips platform for automated and continual in situ monitoring of organoid behaviors, Zhang et al., 2017. DOI: 1073/pnas.1612906114
  • Real-time physiological sensor-based liver-on-chip device for monitoring drug toxicity, Farooqi et al., 2020. DOI:10.1088/1361-6439/ababf4

The manuscript needs at least the characterization of the organoids and the improvement of the sensor integration for a real-time monitoring.

Response: Thank you for your thoughtful advice. Zhang et al. used physical sensors and electrochemical sensors to integrate with liver chips to realize automatic monitoring of PH, oxygen, temperature and protein biomarkers of liver cells. Farooqi et al. microfluid-based electrical and photoelectric sensors continuously measured tra  nsepithelial resistance (TEER) values and pH changes of HepG2 cell lines. However, these sensors are designed for relatively short monitoring time, and do not make full use of the light transmittance of the chip, unable to achieve effective real-time electrochemical and optical monitoring. In this paper, we use electrochemical sensors to monitor cell impedance and near-infrared spectroscopy to measure transmittance and absorbance. After characterizing the performance of the integrated electrochemical and optical sensors, we evaluate the drug response separately.

Reviewer 2 Report

  1. In the references [20] and [21], the authors mentioned two previous works on the electrochemical sensor, however, they should include other works on the near-infrared spectrum studies.
  2. Figure 3 contains no useful information and can be simply removed. Unless the left-hand-side hood picture is removed and replaced with a schematic view of the cell seeding.
  3. line 189, "initially"
  4. line 214; " the value of the two bands were 114 and 35, respectively on the fifth day".

Author Response

Comment 1: In the references [20] and [21], the authors mentioned two previous works on the electrochemical sensor, however, they should include other works on the near-infrared spectrum studies.

Response: Thank you for your significant comment. We have added the description of infrared spectroscopy research, and there are no articles on real-time electrochemical and optical monitoring of liver chips to characterize the performance of liver chips (please see line 59-60).

Comment 2: Figure 3 contains no useful information and can be simply removed. Unless the left-hand-side hood picture is removed and replaced with a schematic view of the cell seeding.

Response: Thank you for your valuable comments. It is true that Figure 3 does not contain any useful information. I have deleted the picture in Figure 3 (please see the revised paper).

Comment 3: line 189, "initially"

Response: Thank you for your careful comments. The first letter of "Initially" on line 189 really needs to be changed to lowercase and I have modified it (please see line 184).

Comment 4: line 214; " the value of the two bands were 114 and 35, respectively on the fifth day".

Response: Thank you for your thoughtful advice. The language of the sentence in line 214 really needs to be modified, and I have revised it (please see line 209-210).